# Gaps in global wildlife trade monitoring leave amphibians vulnerable

**Alice C Hughes[1†]\*, Benjamin Michael Marshall[2†], Colin T Strine[2]**

[1]Centre for Integrative Conservation, Xishuangbanna Tropical Botanical Garden, Xishuangbanna, China; [2]School of Biology, Institute of Science, Suranaree University of Technology, Nakhon Ratchasima, Thailand

**Abstract** As the biodiversity crisis continues, we must redouble efforts to understand and curb pressures pushing species closer to extinction. One major driver is the unsustainable trade of wildlife. Trade in internationally regulated species gains the most research attention, but this only accounts for a minority of traded species and we risk failing to appreciate the scale and impacts of unregulated legal trade. Despite being legal, trade puts pressure on wild species via direct collection, introduced pathogens, and invasive species. Smaller species-rich vertebrates, such as reptiles, fish, and amphibians, may be particularly vulnerable to trading because of gaps in regulations, small distributions, and demand of novel species. Here, we combine data from five sources: online web searches in six languages, Convention on International Trade in Endangered Species (CITES) trade database, Law Enforcement Management Information System (LEMIS) trade inventory, IUCN assessments, and a recent literature review, to characterise the global trade in amphibians, and also map use by purpose including meat, pets, medicinal, and for research. We show that 1215 species are being traded (17% of amphibian species), almost three times previous recorded numbers, 345 are threatened, and 100 Data Deficient or unassessed. Traded species origin hotspots include South America, China, and Central Africa; sources indicate 42% of amphibians are taken from the wild. Newly described species can be rapidly traded (mean time lag of 6.5 years), including threatened and unassessed species. The scale and limited regulation of the amphibian trade, paired with the triptych of connected pressures (collection, pathogens, invasive species), warrants a re-examination of the wildlife trade status quo, application of the precautionary principle in regard to wildlife trade, and a renewed push to achieve global biodiversity goals.

**\*For correspondence:**
ach_conservation2@hotmail.com

[†]These authors contributed equally to this work

**Competing interests:** The authors declare that no competing interests exist.

## Introduction

At the close of a 'decade of biodiversity', we have failed to meet any of the Aichi targets designed to safeguard biodiversity (*CBD, 2020*). One important driver of biodiversity loss is unsustainable wildlife exploitation (*IPBES, 2019*). Countering illegal wildlife trade is critical to limiting biodiversity loss; however, focusing solely on illegal wildlife trade can miss a potentially greater issue: that of legal wildlife trade. Gaps in trade regulations in terms of species covered by international regulation such as by the Convention on International Trade in Endangered Species (CITES) leave groups like amphibians and reptiles among the most frequently traded animals (*Herrel and van der Meijden, 2014*) and largely outside the control of such conventions.

Previous studies aiming to quantify global patterns of trade have relied upon accessible data (such as CITES and IUCN data; i.e., *Scheffers et al., 2019*); relying on regulator data can miss critical legal un/under-regulated trade, as evidenced by analysis on reptiles which highlighted the proportion of species in trade fall outside the scope of CITES (*Marshall et al., 2020*). Such analysis risks providing a false sense of assurance that we understand the dimensions of trade, while in reality the trade may be spanning far more species than those actively monitored (*Marshall et al., 2020*).

**eLife digest** In the last few decades, exotic pets have become much more common. In the UK in 2008, reptiles and amphibians were more popular than dogs, with over eight million in captivity. But while almost all pet cats and dogs are born and bred in captivity, exotic pets are often taken from the wild, putting species and their habitats at risk.

An international trade agreement called the Convention on International Trade in Endangered Species (CITES) strives to prevent unsustainable animal trade. But to get CITES protection, species depend on data showing that wildlife trade threatens their survival. In addition, their range countries need to first propose them to be listed. For most wild animal species, there are no data on population size or population decline. In the case of amphibians, CITES regulates the trade of just 2.5% of species. This leaves the rest with no protection from overarching international trade regulations. To protect these animals, researchers need to find out which species are in trade, where they are coming from, and how many are already threatened.

To address this, Hughes, Marshall and Strine combined data from five sources, including official CITES trade records, recent research and an online search for amphibian sales in six languages. The data showed evidence of trade in at least 1,215 amphibian species, representing 17% of all amphibians. The figure is three times higher than previous estimates. Of the species in trade, more than one in five is vulnerable to extinction, endangered, or critically endangered. For a further 100 of the traded species, data on population were unavailable. Moreover, analysis of the origins of traded individuals showed that around 42% came from the wild. Tropical parts of the world had the highest number of species in trade, but the data showed exchanges happening across the globe.

Unsustainable wildlife trade can have devastating consequences for wild animals. It has already driven at least 21 reptile species to extinction, and data of amphibian species are unknown. To prevent further species going extinct, legal wildlife trade should follow the precautionary principle when it comes to wildlife trade. Rather than allowing people to trade a species until CITES regulates it, a blanket ban should come into force for species that have not been assessed or are threatened. Trade would be able to resume for a species only when assessments show that it would not cause major population decline, or secure, captive breeding facilities can be guaranteed.

---

*Marshall et al., 2020*, highlighted the discrepancy in protection within the reptile trade, with only 8.3% under CITES regulations yet over 36% in trade and over 70% of individuals from some taxa (e. g., lizards) harvested from the wild (*Marshall et al., 2020*; *Uetz et al., 2021*). Whilst trade of wild-collected individuals is not necessarily unsustainable, such a judgement should rely on data, as unregulated harvest from the wild, especially for rare or small-ranged species could potentially pose a significant risk to the continued survival of such populations (*Auliya et al., 2016*).

The need for a complete assessment of amphibian species in trade, their origins, and where native populations are at risk is emphasised by targeted studies revealing high rates (87% of individual Southeast Asian newts) of wild collection (*Rowley et al., 2016*). Given that species can be restricted to single drainage basins, unsustainable trade can represent a genuine risk to species future survival; limited trade assessments means that understanding when trade is or is not sustainable simply is not possible for many species, though recent studies show it can have an impact on population viability (*Morton et al., 2021*).

Despite experiencing similar pressures to reptiles and greater sensitivity to perturbations (*Stuart et al., 2004*), amphibians are one of the least protected taxa under CITES regulation with only 2.4% of all known species listed (second only to fish at 0.46%: http://www.fishbase.org/home.htm), despite showing faster population declines than any other vertebrate group (*Hoffmann et al., 2010*). Often dubbed *canaries in the coal-mine* amphibians are sensitive to a myriad of anthropogenic stressors: pollution (*Blaustein et al., 2003*), habitat loss (*Stuart et al., 2004*), atmospheric changes (*Blaustein et al., 2003*), introduced pathogens (*Lips, 2016*), invasive species (*Bellard et al., 2016*), wildlife collection (*Phimmachak et al., 2012*), and agricultural chemicals (*Trudeau et al., 2020*); such stressors are exacerbated by amphibians' frequently small distributions and naturally fluctuating populations (*Nori et al., 2018*; *Luo et al., 2015*; *Hu et al., 2012*). Amphibian trade is directly tied to the last three stressors. Trade can enable pathogen spread (*O'Hanlon et al., 2018*),

which has facilitated devastating amphibian species loss (*Scheele et al., 2019*; but see *Lambert et al., 2020*, for concerns over the number of species). Invasive amphibians (often linked to trade; *Lockwood et al., 2019*; *Stringham and Lockwood, 2018*) can be vectors for pathogen spread (*Bienentreu and Lesbarrères, 2020*; *Feldmeier et al., 2016*), but also can compete with native species for resources such as space and prey (*Falaschi et al., 2020*). Wild collection (directly taking animals from the wild) occurs at several scales: on local levels, humans collecting species for trade, consumption, and medicine (*Ribas and Poonlaphdecha, 2017*; *Van Vliet et al., 2017*; *Onadeko et al., 2011*), whereas more widely amphibian trade is augmented by demand for pharmaceutical products, pets, and even fashion (*Auliya et al., 2016*; *Xiao et al., 2011*).

A recent literature assessment of amphibian pet trade found 443 traded species (*Mohanty and Measey, 2019*), but as we strive towards ever more complete and representative assessments of the amphibian trade, we must capture trade other than pets, as well as outside of literature (that can often be skewed towards certain languages/regions; *Konno et al., 2020*). More standardised and comprehensive data are necessary to ensure that wildlife trade avoids harming species' long-term survival prospects; the current lack of data and thus lack of transparency or access to baseline population data and compiled trade records frustrate trade mitigation efforts.

Here, we aim to map amphibian species in trade, complementing previous regional efforts (*Yap et al., 2015*), or those focusing on easily accessible data such as CITES (CITES trade database; https://trade.cites.org) and LEMIS (United States Fish and Wildlife Service's Law Enforcement Management Information System). We explore two major inventories of international trade, combining this with an automated web search of amphibian selling websites across six languages. We place these findings in the context of the findings of the *Mohanty and Measey, 2019*, and species reported as traded within the IUCN Redlist species assessments. In addition, we examine the overlap between these five trade data sources and explore the different trade dimensions they represent, and how the trade may impact wild populations. We further explore where species origins and their threat status, thereby attempting to highlight trade vulnerability hotspots. This study builds towards a comprehensive assessment of amphibian trade, while attempting to highlight how many species are traded, the major drivers of trade, and where these species originate.

## Results

We split our assessment of the trade into contemporary trade and all trade. Contemporary trade used three trade inventories which could be examined for trade dynamics (LEMIS 2000–2014, CITES 1975–2019, and Online trade 2004–2020). All trade also included two additional datasets reporting species presence in trade (IUCN Redlist species assessments; *Mohanty and Measey, 2019*).

### Dimensions of trade

Our online search efforts successfully examined a total of 139 amphibian selling websites and retrieved 2766 web pages to be searched (mean of 19.91 ± 3.95 pages per website; range 1–302). Our temporal online sample (2004–2019) added an additional 4568 pages, meaning our complete online species list is based on searches across 7334 pages in total. We detected 480 keywords (i.e., amphibian scientific names and synonyms) that equated to 442 species in the 2020 snapshot, and 486 keywords that equated to 443 species in the temporal sample, resulting in a total of 575 species detected in the Online trade.

Overall, the three data sources (Online trade, LEMIS, and CITES trade database) contained 909 species in total (11.06% of the 8212 total described amphibian species), of which LEMIS had the most (587 species, 31% unique), followed by Online trade (575 species, 30% unique) then CITES (137 species, 4% unique). Most of this trade was commercial (99.6%) with only 0.4% non-commercial. Unsurprisingly, anurans (729 species) dominated the trade, followed by salamanders (162 species) and caecilians (18 species). Based on these three trade inventories, a total of 157 species were threatened (i.e., listed as Vulnerable (VU, EN, CR) or worse on the IUCN Redlist), 27 Data Deficient, and 39 unassessed, and the remainder Least Concern (*Figure 1*).

Whilst the majority of species in trade in CITES have a CITES appendix (95%), this is not the case for species detected via LEMIS (14%) or online searches (16%). In terms of the degree of threat, 47% of species in trade via CITES are threatened according to the IUCN and 12% are unlisted by the IUCN, whereas this is lower for LEMIS (24%; 5%) and Online (23%; 6%). However, due to the larger

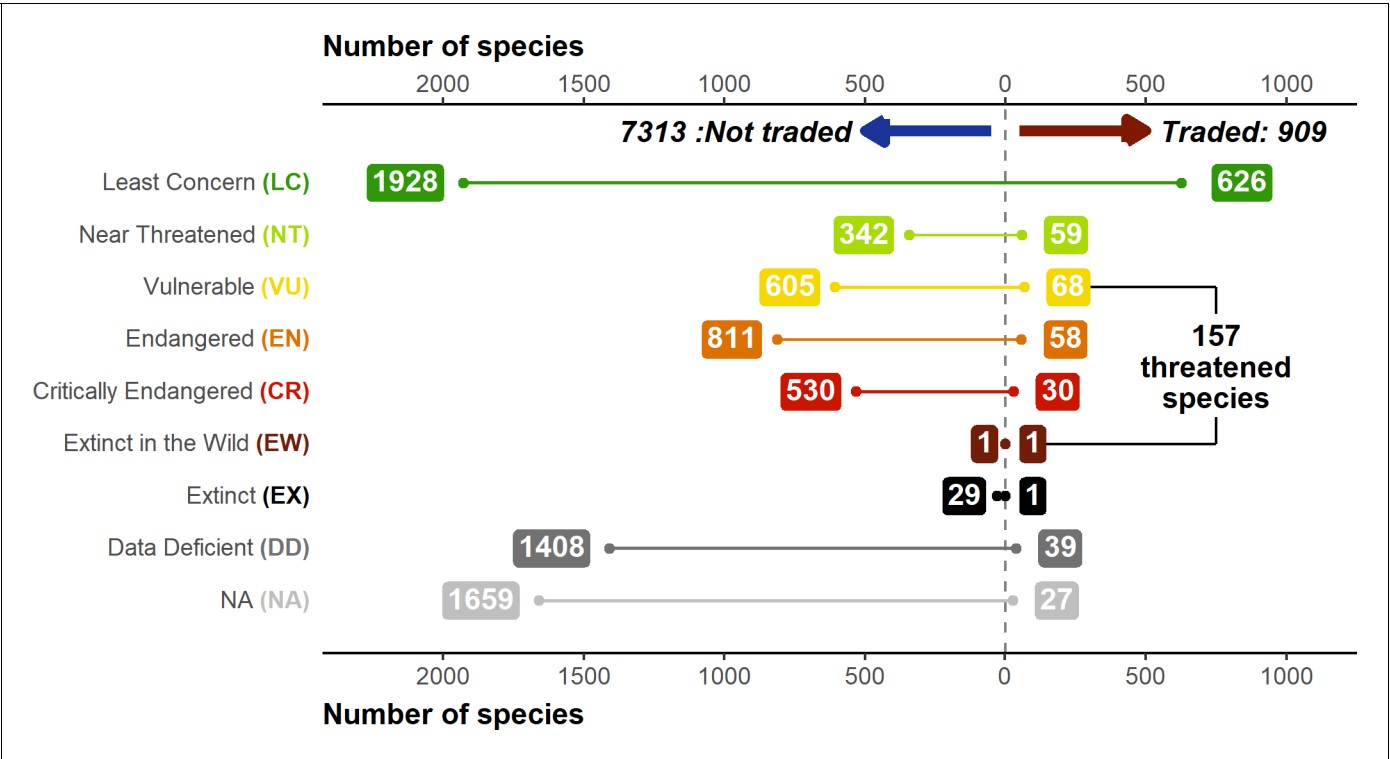

**Figure 1.** Breakdown of IUCN Redlist status of traded and not-traded amphibian species. IUCN assessments based on data from AmphibiaWeb. Inclusion as a traded species based on appearance in online searches (2004–2019 and 2020 online contemporary sample), Law Enforcement Management Information System (LEMIS) (2000–2014), and Convention on International Trade in Endangered Species (CITES) data sources (1975–2019). Generated using *Source code 8* and *Source data 10*.

number of species traded, species detected via LEMIS and online searches account for a larger proportion of all threatened amphibian species. For example, 4% of Critically Endangered species and 5% of Endangered species were detected in trade via LEMIS, compared to 2% and 3% for CITES. In total, relying exclusively on CITES would suggest only 3% of threatened species are traded, whereas LEMIS and Online reveal 5% of threatened species traded, with most threatened species in trade not listed by CITES.

Mapping reveals a global exploitation of amphibians. However, the number of species exploited in different regions varies dramatically (*Figure 2*; *Figure 2—figure supplement 1*). Both LEMIS and Online trade highlight high numbers of species across the tropics, especially in the Amazon. However, LEMIS highlights more traded species in Africa and Southeast Asia, and CITES misses almost all areas with only a fraction of species in the Amazon (poison dart frogs) covered (*Figure 2—figure supplement 2 and* 3). Particularly high proportions of species were in trade, not only in less diverse regions, but also across tropical Asian regions. In addition, particularly high percentages of species are in trade in South Cambodia and areas of Madagascar (*Figure 2—figure supplement 2 and 3*).

Many traded species categorised as Vulnerable or worse originate from East and Southeast Asia, in addition to the Mediterranean and various parts of South America (*Figure 2—figure supplement 2 and 4*), whereas small-ranged species are in trade from across the tropics and various islands. At the national level, countries across the Middle East and Southeast Asia had more than half their species in trade classed as either threatened or Data Deficient/unassessed. South America, Madagascar, and the Caribbean have even higher percentages of threatened species in trade. South America and Southeast Asia have the highest numbers of species in trade without CITES regulations.

The LEMIS trade inventory provides us with greater insights into the source of the amphibians being traded. Of the trade described in LEMIS 2000–2014, and constituting/representing single individual animals, 99.9% is not from seizure and enters the USA (69,688,337/69,771,677), and the vast majority is for commercial purposes (69,492,478/69,771,677; 99.6%). Of the 69,771,677 amphibians

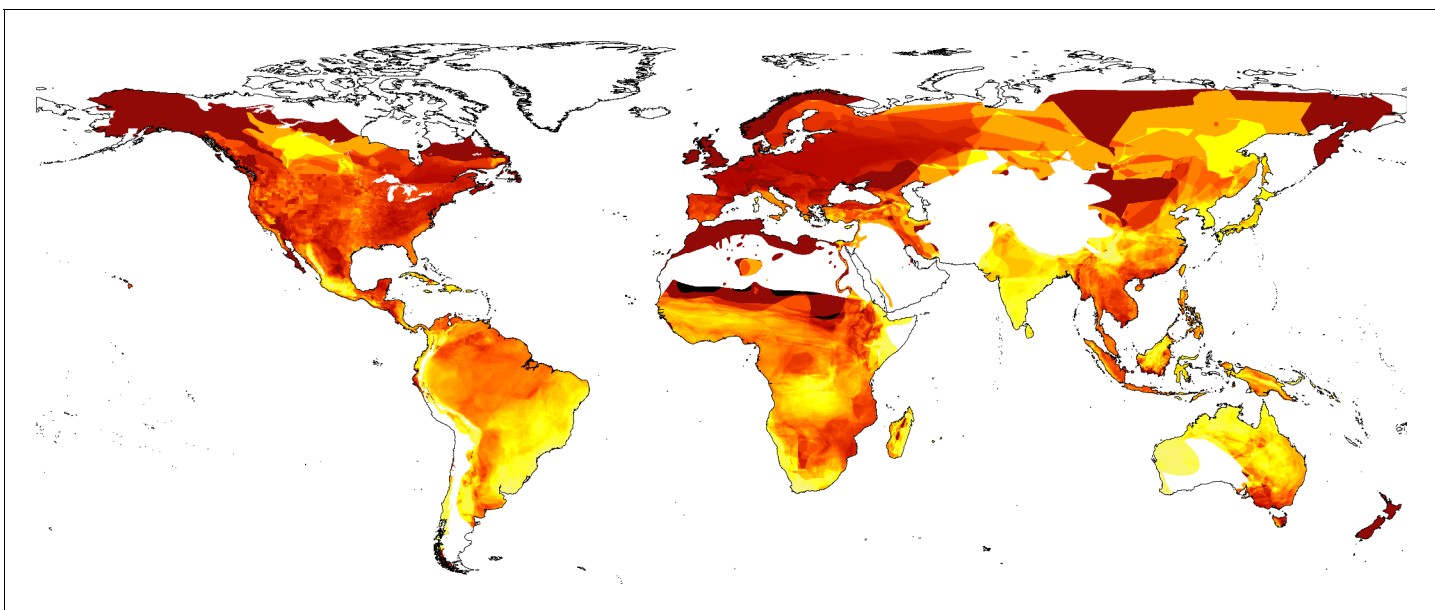

**Figure 2.** Percentage of species in trade based on three combined contemporary datasets (Law Enforcement Management Information System [LEMIS], Convention on International Trade in Endangered Species [CITES], Online [yellow (0%)-red-black (100%)]). Also see *Figure 2—figure supplements 1*, *2*, *3*, and *4* for patterns of individual countries and inventories.

The online version of this article includes the following figure supplement(s) for figure 2:

**Figure supplement 1.** Map of trade by country derived from Online, Law Enforcement Management Information System (LEMIS), and Convention on International Trade in Endangered Species (CITES) trade data, and mapped using AmphibiaWeb distribution data.

**Figure supplement 2.** Species traded from different trade inventories.

**Figure supplement 3.** Maps of national statistics of species with different IUCN.

**Figure supplement 4.** Maps of threatened species in trade based on the three trade inventories.

imported into the USA, recorded by LEMIS, 57.2% (39,921,289) are listed as captive sourced, leaving 42.3% (29,522,128) as originating from the wild (the remaining 0.47%, 328,260, classed as other or with an ambiguous source). The wild capture volumes and percentages vary among genera, from millions of individuals to fewer than 100 (*Figure 3—figure supplements 1–6*). The vast majority of imported genera are impacted by wild capture (254/259) with 141 genera exclusively wild-sourced; five genera are fully sourced from captive operations (*Peltophryne*, *Ranitomeya*, *Calyptocephalella*, *Cryptophyllobates*, *Samandrella*; *Figure 3—figure supplements 1–6*). On average 84.2% of each genera's individuals come from the wild, and a per genera median of 100% is likely driven by the large number of genera exclusively taken from the wild but in much lower volumes (e.g., fewer than 100 individuals, or fewer than 10 individuals per year given the 2000–2014 timeframe; *Figure 3—figure supplement 6*).

## Trends over time

Whilst the CITES trade has remained relatively consistent over time between 2000 and 2020 at around 50 species a year with a gradual increase of species, LEMIS shows an increase up to 2014 (the limit of available data) at 310 species (*Figure 3A*). The Online trade shows much more interannual variation (likely exaggerated by sampling effort fluctuations), increasing to 200 species in 2010, decreasing up to 2014 at under 100 species, then increasing again up to over 200 species in 2019. The number of pages scraped for online trade also followed this trend, peaking at over 1250 pages in 2014, decreasing to under 200 in 2014 then increasing to over 1000 in 2018 (*Figure 3B*). The residuals from a linear regression accounting for the number of pages searched suggests a steady increase in species (*Figure 3B*).

Thirty-eight species described since 1999 (1.38% of the 2747 amphibian species described after 1999; *Figure 4A and B*) appeared in trade based on our three inventories (and 41 with the addition

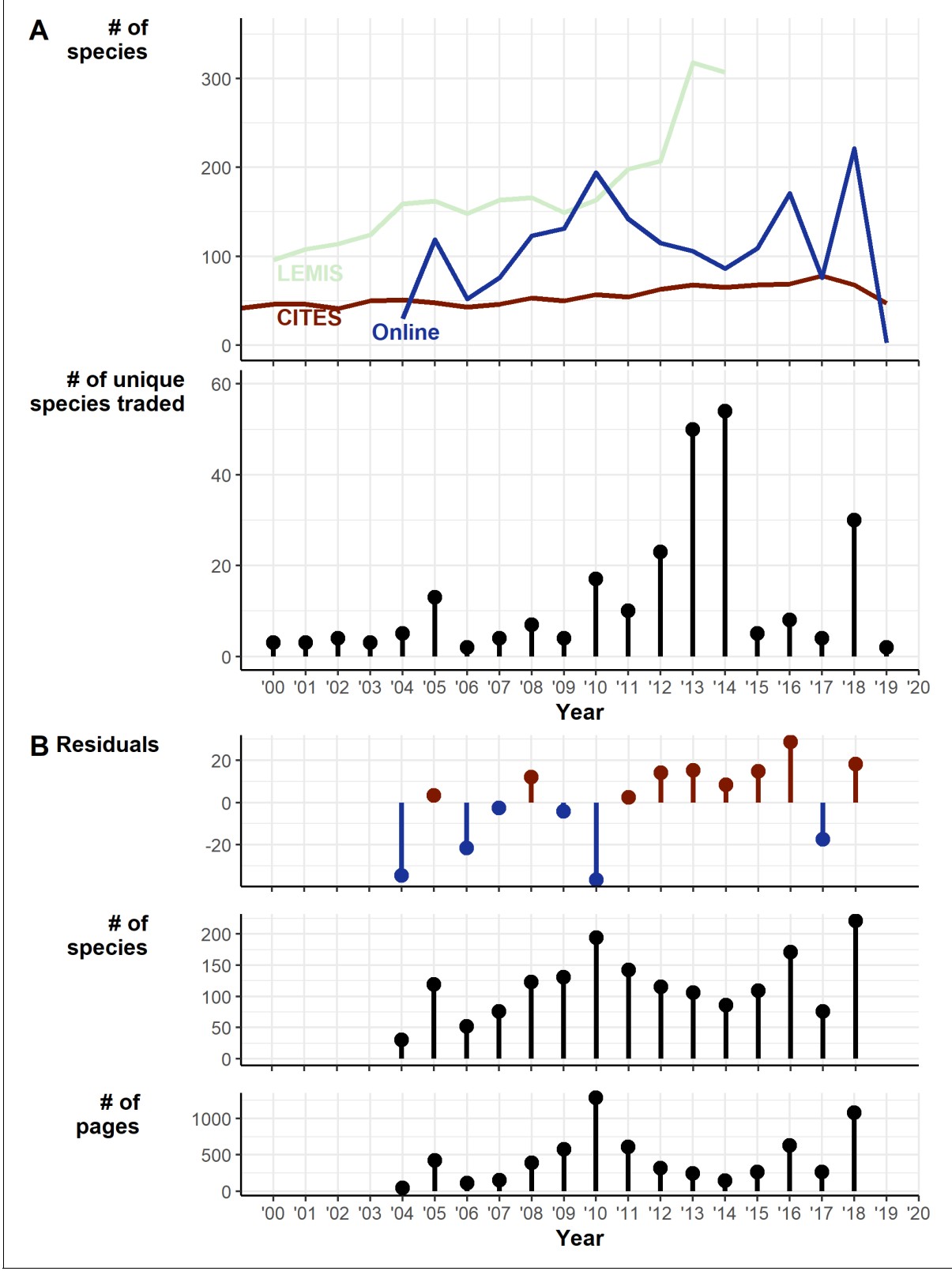

**Figure 3.** Temporal trends in traded species 2000–2019. (**A**) Trends over time of Online, LEMIS, and CITES datasets: (1) Raw counts of numbers of species detected in each year. (2) The number of species traded only in a particular year. (**B**) Exploration of trends in online trade: (1) Residuals from the linear regression of number of species detected against number of pages (df = 13, intercept = 58.73, number of pages coef. = 0.13). (2) Number of

*Figure 3 continued on next page*

*Figure 3 continued*

species per year. (3) Number of archived pages retrieved and searched. Generated using *Source code 9* and *Source data 7*, *9,* and *10*. Also see *Figure 3—figure supplements 1–6* for a breakdown of how many individuals are coming from the wild for taxa traded at different volumes. The online version of this article includes the following figure supplement(s) for figure 3:

**Figure supplement 1.** Bar chart showing the number and origin of imported individuals per genera, subset to genera with over 1,000,000 individuals recorded.

**Figure supplement 2.** Bar chart showing the number and origin of imported individuals per genera, subset to genera with between 1,000,000 and 100,000 individuals recorded.

**Figure supplement 3.** Bar chart showing the number and origin of imported individuals per genera, subset to genera with between 100,000 and 10,000 individuals recorded.

**Figure supplement 4.** Bar chart showing the number and origin of imported individuals per genera, subset to genera with between 10,000 and 1000 individuals recorded.

**Figure supplement 5.** Bar chart showing the number and origin of imported individuals per genera, subset to genera with between 1000 and 100 individuals recorded.

**Figure supplement 6.** Bar chart showing the number and origin of imported individuals per genera, subset to genera with fewer than 100 individuals recorded.

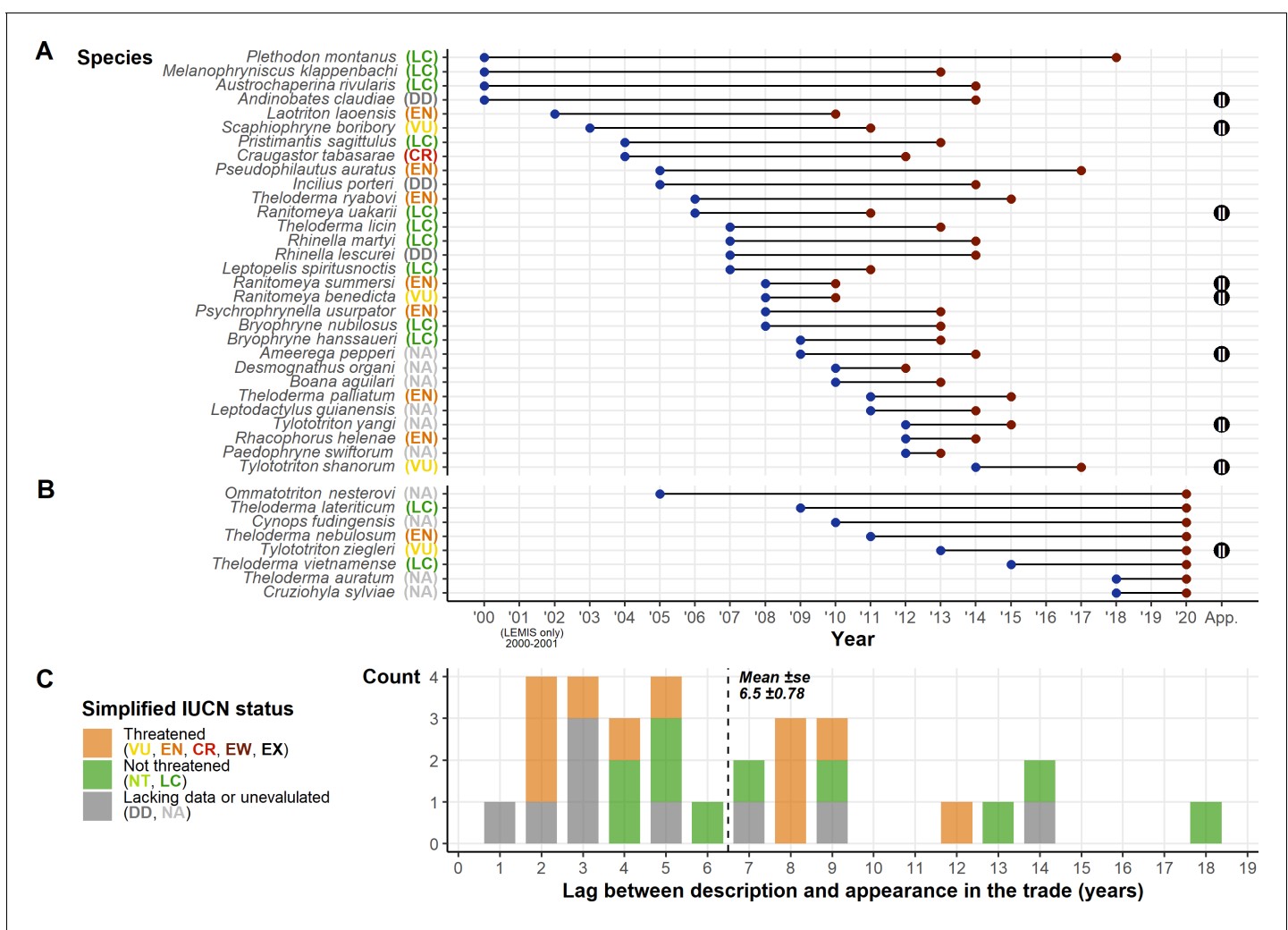

**Figure 4.** Summary of post-1999 described species and their presence in the trade. (A) The species described post-1999 detected in the trade displaying the year of description and the year detected in the trade. (B) Species described post-1999 but were only detected in the 2020 snapshot. Alongside species names in A and B are their IUCN Redlist status; the Convention on International Trade in Endangered Species (CITES) appendix (where listed) is shown on the right of the plot. (C) Frequency plot showing the count of time lags between description and trade, with colours corresponding to broad summaries of IUCN Redlist status. Generated using *Source code 11* and *12*, and *Source data 4*, *7,* and *10*.

of two further species described in 2018 and listed for sale in 2020; *Altherr and Lameter, 2020*). Eight only appeared in the 2019 snapshot, so are discounted from time lag calculations, leaving 30 species with connected trade years and a mean time lag of 6.5±0.78 years between species description and appearance in the trade. Of the 38 species, 12 are Least Concern, 10 are unevaluated, three are Data Deficient, and 13 are threatened (one of which is Critically Endangered). One species was in trade the year after it was described, but four were in trade in the second year, four in the third year, and seven within 4–5 years (*Figure 4C*). We cannot differentiate instances of rapid exploitation after species description from instances of name updates pertaining to species already traded. Although it should be noted that even in these cases given the smaller population sizes and distributions of split species, they may be more vulnerable to population declines resulting from wild-harvest, as populations and ranges are likely to be smaller than currently known.

## Language markets

Different language searches returned different species lists, with all languages containing species unique to that language. English and German detected the most species by far (293, 289), and each also contained the highest rates of unique species (81, 97). German produced a larger list of species, despite similar sampling efforts as Spanish, French, Japanese, and Portuguese (*Figure 5*). The top websites in terms of species were mostly commercial (six out of the top ten), two of which prominently advertised wholesale options. The remaining four top websites (including the top website with 278 species) were hosting classified advertisements.

## Drivers of demand

To better capture all the species traded, we combined our contemporary analyses from the three data sources (Online trade, LEMIS, and CITES trade database) with the analyses from *Mohanty and Measey, 2019*, and the IUCN Redlist assessments. Comparisons reveal that different sources detected different species in the trade, and no single source is sufficient to detect all species traded (*Figure 6*). Combining all sources yielded a total of 1500 amphibian species in trade before

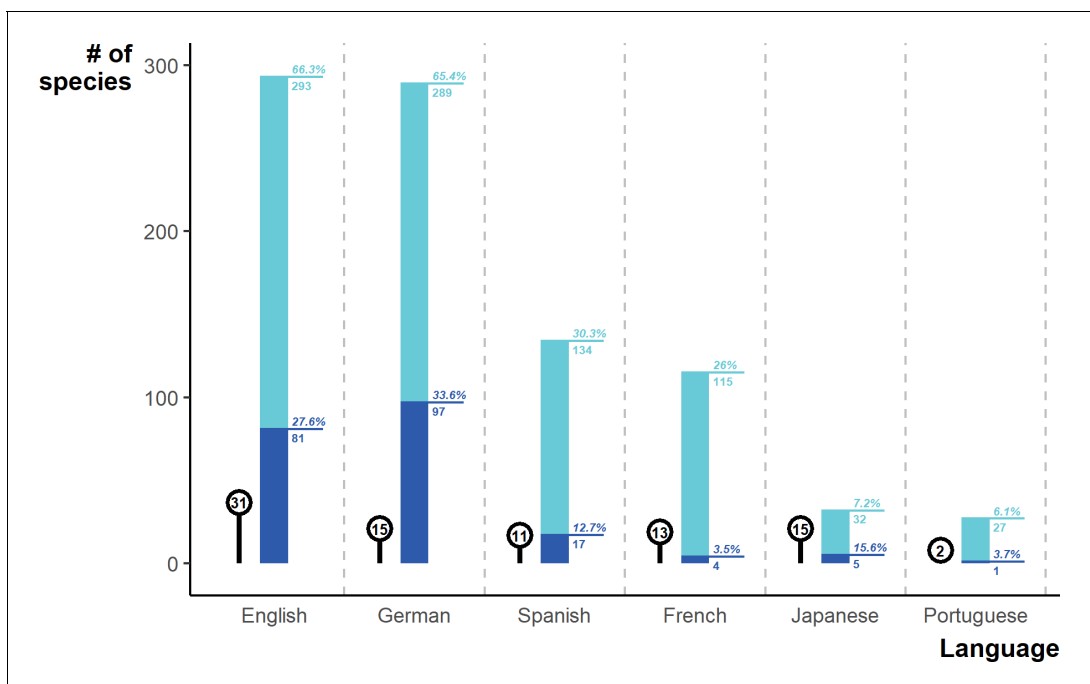

**Figure 5.** Number of species detected via each language in the online search. Light blue shows the total number of species per language, and percentage of the overall online species list. Dark blue shows the number of species unique to a particular language and the percentage of that language's species that are unique. Lollipop alongside bars describe the number of websites sampled. Generated using *Source code 10* and *Source data 1* and *3*.

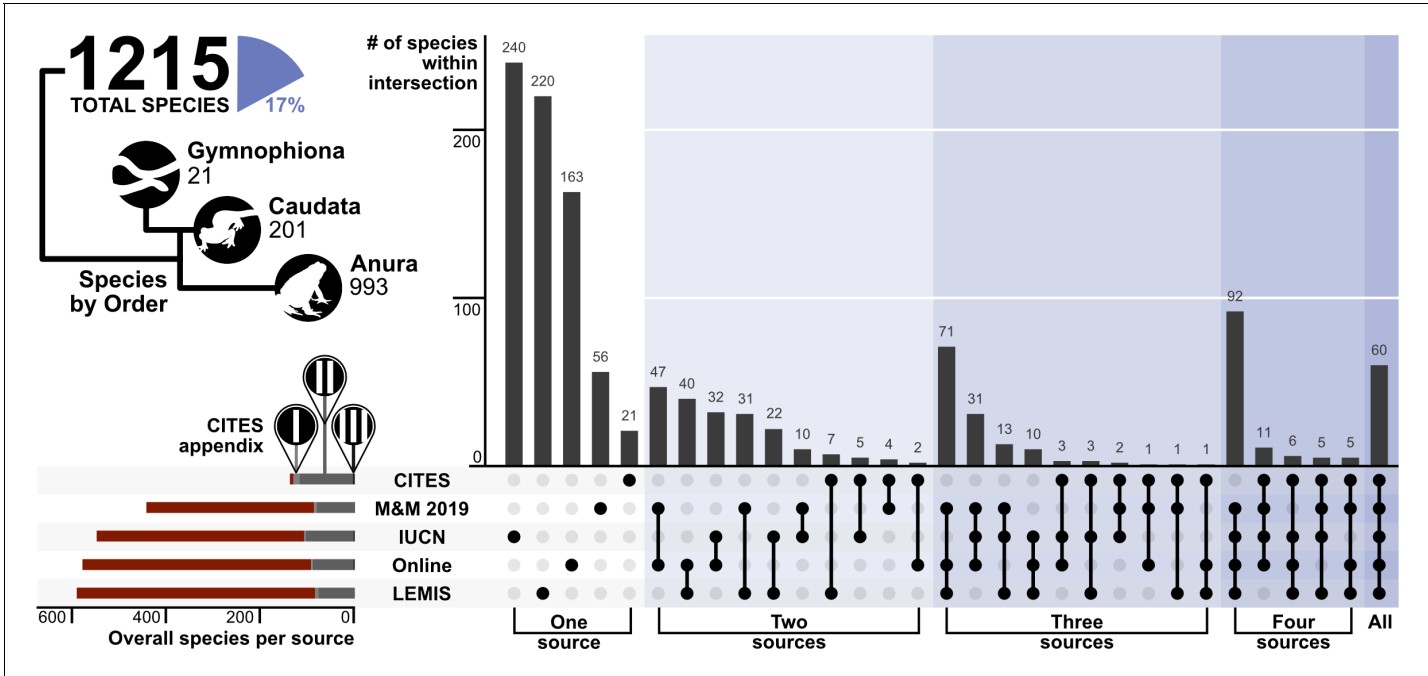

**Figure 6.** Upset plot showing the coverage and intersection of the five trade data sources. The number of species per order is presented as an illustrative tree, alongside the % of the 8212 amphibian species in trade. The number of species that are covered by each CITES appendix is represented in the bottom left plot (red – not listed, light grey – Appendix I, medium grey - Appendix II, black – Appendix III). N.b., M&M 2019 is referring to *Mohanty and Measey, 2019*. Generated using *Source code 8*, and *Source data 10*.

synonyms were removed, and 1215 once synonyms were removed, equivalent to 17% of amphibian species.

The 1215 species included up to 413 species used for meat (though a significant number were largely local consumption based on IUCN assessments), 805 species as pets (though six are from separate lists: one from Germany; *Altherr and Lameter, 2020*; five from Asia; *Choquette et al., 2020*), 122 species used as medicine or in pharmacological research, and 664 species imported for research or breeding facilities (including zoos and aquaria); other purposes were also listed (various fashion companies such as Prada and Gucci were listed as importers, and some amphibians are imported for bait) but we have not listed these uses separately. In total over 930 species were used for other commercial purposes, and 1215 species in total when medicinal/pharmaceutical and research are included. In terms of status, 4% of species in trade are Critically Endangered (4% for pets, 4% for meat), 10% are Data Deficient or unassessed (9% pets, 11% meat, over 8% used in medicine or pharmacology). In total, 22% of species in trade are threatened (i.e., Vulnerable or worse, 28% when Near-Threatened are also considered), 25% for pets, 31% for meat, 39% for medical purposes and only 21% of those used for research. In terms of coverage of species for each type of trade by CITES (12% overall 151/1215), this varied from 5% of species used for meat, to 16% of those used for pets or 18% for medicine, and 16% of those in research.

Mapping out these patterns also revealed a variety of trends among different uses (*Figure 7*). In terms of commercial trade, pet trade dominated the global trade of amphibians and the pattern is most similar to the map of all trade with up to 51 species from any given area shown to be in trade for pets relative to the 71 from all trade. Trade for meat is more limited with only up to 26 species from any given area in trade, and up to eight species for medicine or pharmaceutical trade. Interestingly, research/zoos were associated with up to 57 species from any given area in trade and broadly mirroring the patterns seen in the pet trade. It should be noted that these may be underestimates, as the LEMIS does not specify exact purpose, and it must be inferred from the buyer and type of sale. Whilst the volumes likely differ substantially between animals traded for research relative to commercial sources, it highlights the numbers of species potentially vulnerable to at least low levels of international trade. Commercial trade of amphibians for meat is also shown to be from Asia using

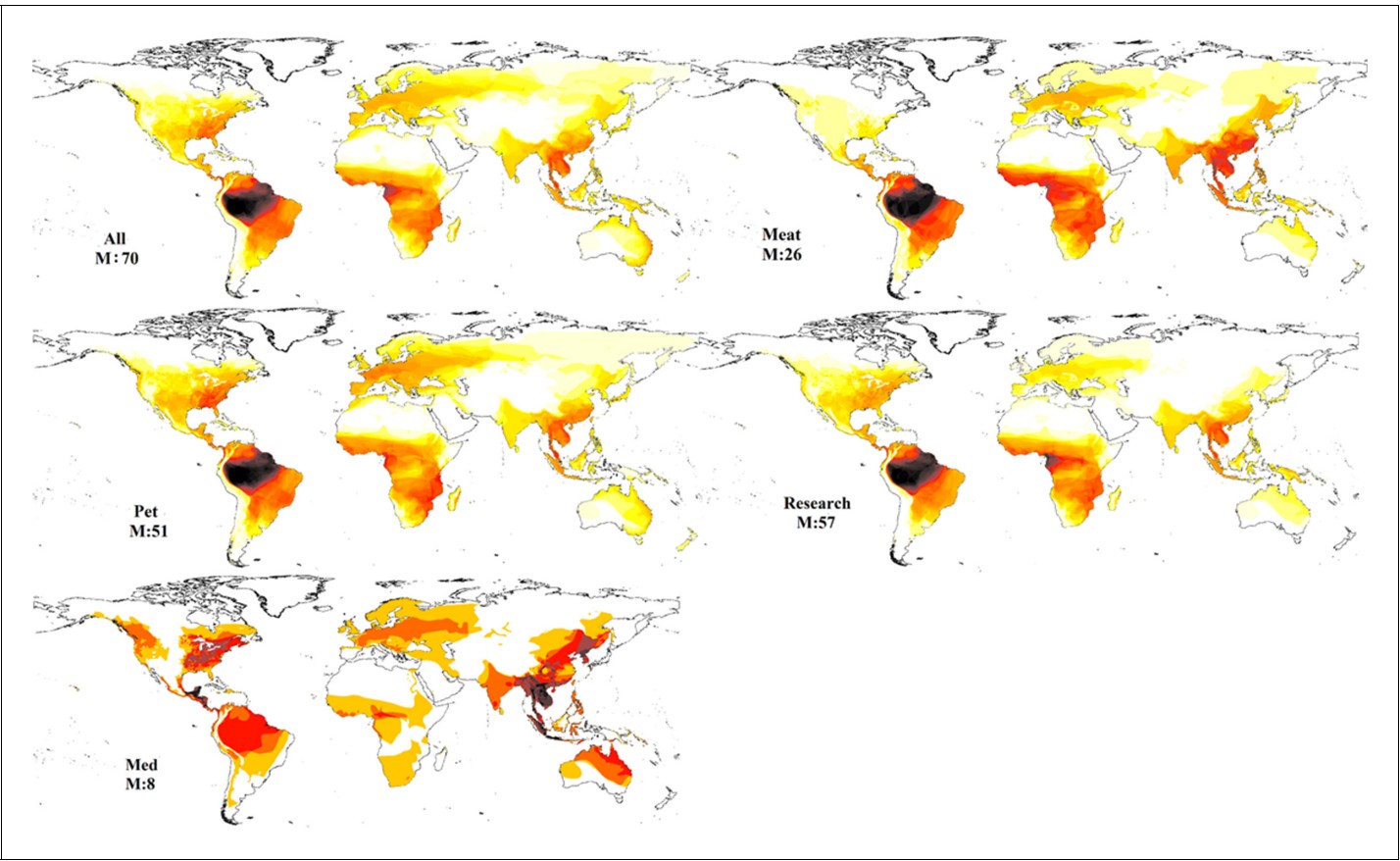

**Figure 7.** Mapping diversity of species in trade for different uses based on the five data sources. (**A**) Pet, (**B**) meat; (**C**) medicinal, (**D**) research, and (**E**) all trade.

the United Nations Commodity Trade Statistics Database (UN Comtrade: https://comtrade.un.org/data/) which shows that global export of frog legs is dominated by Indonesia (at 8,005,997 kg in 2008–2009 alone), followed by China, Vietnam, and other Asian nations with the dominant markets in France, Belgium, and the USA, though these statistics are only available until 2010 and markets seem to be both growing and diversifying at that point, based on data available in the preceding years.

## Discussion

### Scale, scope, and vulnerability

Amphibian declines are often considered to provide an early warning of potential declines in other taxa as they are sensitive to pollution and habitat loss making their absence an early warning sign of habitat degradation; sensitivity to change combined with trade, and disease risk creates the perfect storm threatening future amphibian survival.

Whilst regional and some global studies have explored the extent of pet trade (*Measey et al., 2019*), or meat trade (*Carpenter et al., 2014*), a well over double the known number of species are in trade relative to previous studies (i.e., *Scheffers et al., 2019*, 542 relative to 1215), as well as a more representative understanding of what is currently in trade and how it has changed over the last two decades. The scope of the amphibian trade is larger than formerly realised with implications for the direct exploitation of these species, disease spread (*Schloegel et al., 2009*), and the pool of potentially new invasive species (*Gippet and Bertelsmeier, 2021*). Each dataset we examined included unique species missing from the other datasets (*Figure 6*), illustrating the need to use

multiple sources to characterise wildlife trade, and underscoring the need for a better system to centralise knowledge on what is being traded, and where animals are sourced.

Concerns over the scale and scope of the trade are compounded by the lack of baseline population studies, frustrating efforts to truly understand sustainability of the trade, as understanding sustainable offtake is impossible without baseline population data. A recent meta-analysis on how trade impacts wild populations was unable to generate an estimate on amphibians because of a lack of standardised studies, but revealed abundance declines of 62% (95% CI 20–82%) in traded populations of mammals, birds, and reptiles (*Morton et al., 2021*). Amphibians in areas with high volumes of exports may be at particular risk given the high rates of wild capture. For example, meat trade is known to impact at least 40 species annually from Indonesia alone (*Gratwicke et al., 2010*), with many coming directly from the wild, and even captive rearing facilities risk endangering wild species through pathogen exposure unless biosafety standards are improved. Understanding the impacts of harvest and trade on source populations requires a better understanding of what species are being traded, the volumes in trade and the status of the wild populations is critical for preventing negative impacts on source populations, especially given that the IUCN assessments can be decades old and not accurately reflect species' current threat status (*Natusch and Lyons, 2012*). Furthermore, quantitative analysis of the volumes of species in trade often relies on import data (e.g., LEMIS) and ignores mortality during transit and transport, which has been shown to be as high as 72% in some studies (*Ashley et al., 2014*), with mortality in amphibians higher than all other groups (45% within 10 days of confiscation). Such statistics are alarming, and also highlight that the number of animals exported may be far higher than the anticipated demand to compensate for mortality before sale.

Despite the impact of trade, the World Customs Organization still fails to list species data in exports – only basic data is needed to legally export most amphibians, providing no species-specific information to enable trade monitoring. With limited baselines on populations and disparate or inaccessible records of trade, we cannot hope to make effective management decisions or develop quotas and tools for sustainable use. A lack of systematic monitoring of global trade limits us to a basic understanding of traded species, origin, and impacts on native species. Monitoring deficiencies have been repeatedly highlighted over the past decade, but we still await the policy responses necessary to ensure the survival of vulnerable species (*Auliya et al., 2016*). In fact, government funding for projects targeting basic monitoring initiatives has dwindled in recent years in favour of applied scientific applications, and 'less charismatic' species are most likely to be underfunded (*Bellon, 2019*) and have lower investment in conservation (*Gerber, 2016*).

We show 22% of the 1215 species in trade are threatened (i.e., IUCN Redlist status of Vulnerable or worse), and a further 8% remain unassessed or Data Deficient. One in ten traded species are already highly threatened (11% of species Endangered or Critically Endangered). The trade extends beyond captive-reared or ranched individuals, and is motivated in part by novelty and rarity (as has been documented for the reptile trade previously; *Marshall et al., 2020*; *Lyons and Natusch, 2013*), potentially further illustrated by the appearance of 38 species described since 2000 in the trade. Whether these new species are the result of species splits or completely novel lineages being described, they highlight the knowledge gaps that need to be addressed before sustainability can be confidently assessed. However, *Stringham et al., 2021*, showed that new (reptile) species smuggled in Australia were well predicted by their existence in US markets, thereby suggesting a diminished role for novelty (i.e., recent description) when compared to accessibility. Because of novelty dynamics in trade and changing taxonomy, CITES appears an inadequate tool to describe taxonomic or spatial trade patterns; CITES does not include 97.5% of amphibian species, and fails to provide any default (or sufficiently rapid) protection for newly described and potentially vulnerable species, and even scientific descriptions of species have been found to enable these newly described species to be targeted for trade (*Yang and Chan, 2015*; *Yeager et al., 2020*). Tropical regions and islands, with high levels of endemism, still have a significant proportion (often exceeding one-third or even half) of species traded indicating the need to expand trade monitoring, and to prevent trade as a default until non-detriment findings can be assessed for any potential trade.

Global monitoring continues to be inadequate; the lack of specificity hinders the utility of global data from the World Customs Organization (*Chan et al., 2015*). Calls for improvements and increased specificity were made at the IUCN's 5th World Conservation Congress (WCC-2012-Res020) in 2012. Changes remain elusive, with details on updates in the *World Customs Organization, 2020*, edition failing to address animal trade (*World Customs Organization, 2020*). Thus, a

decade has passed and reasonable actions for the conservation of biodiversity are still ignored in economically orientated databases. The dearth of reliable/accessible data (both for baseline population and trade volumes) undermines efforts to determine trade sustainability for the vast majority of non-CITES species (i.e., the vast majority of all amphibian species). The trade of Endangered and range-limited species, paired with the high rates of wild capture (especially given that this is higher for pets than for other purposes), would suggest much of the trade could be unsustainable and damaging the future survival of species.

## Trade and disease

To date, 94 cases out of the 159 extinct and potentially extinct species from the 2008 Global Amphibian Assessment are at least partially attributed to *Batrachochytrium dendrobatidis* (Bd) (*MacCulloch, 2008*; *Picco and Collins, 2008*), and suggestions that Bd is likely to be responsible for up to 500 species declines (*Scheele et al., 2019*; but see *Lambert et al., 2020* for discussion on the 500+ estimate). Furthermore Bd, *Batrachochytrium* salamandrivorans (Bsal)*, Ranavirus* and a range of other diseases, carried by amphibians and fish, can spread into naïve populations and move between aquatic taxa (*Bayley et al., 2013*; *Mao et al., 1999*; *Densmore and Green, 2007*). With millions of individuals exported annually (peaking at around 5575K kg from Indonesia alone in a single year in the early 1990s, and fluctuating between 3600K and 5000K kg most years based on LEMIS), no systemic mechanism to ensure correct identity, and poor biosafety standards, water contamination resulting from continued unrestricted/uncontrolled trade is likely to lead to further disease spread, and population declines. Rates of Bd in live exports can be high (over 60% of individuals), with studies linking the spread of Bd and Bsal to the trade of live animals in the pet trade (*Fitzpatrick et al., 2018*; *Kriger and Hero, 2009*; *Yuan et al., 2018*). As a consequence of this risk of disease, areas like the European Union have initiated the TRACES (TRAde Control and Expert System) programme to attempt to monitor what is imported and associated disease risk. Yet, such data is challenging to access and is unlikely to enable proactive monitoring for ecosystem health, despite the development of organisations such as the World Organisation for Animal Health (OIE) (*Martel et al., 2020*). However, regional networks have been developed for specific cases such as Bd such as spatialepidemiology. net (Aanensen, 2009).

The risk of both recognised and novel invasive pathogens should not be underestimated. Whilst we did not separately map it here, various amphibians are sold commercially as bait. Previous studies show that not only do the live animals kept in bait shops frequently carry fungal and viral pathogens, but they are also frequently released into the wild after use (*Picco and Collins, 2008*). Given that over 40% of individuals in this study are shown to come directly from the wild, the potential for spread of pathogens to spread to new areas must be addressed to avoid severely impacting native aquatic vertebrate communities (*Price et al., 2017*).

## The necessity for change

Many papers have highlighted the inadequacies of a CITES paper-based system for monitoring trade (*Berec et al., 2018*). In the context of amphibians, the discrepancies on reporting (such as species exported from the wild from countries to which they are not native; *Auliya et al., 2016*) are well documented. Here again, we highlight that CITES fails to provide adequate safeguards both for species which are included, and more so for the 97.5% of amphibian species that are not.

In recent years, millions of amphibians representing over 1200 species have been traded, with a considerable portion of individuals coming from the wild. The trade of range limited, Data Deficient, and newly described species with extremely limited data highlights how harm to species future survival prospects may be occurring out of sight. Inadequate biosafety standards, potential escape, and invasive species in combination with the direct exploitation threaten the future survival of species. The World Customs Organization must urgently address the lack of coding for these species, to enable steps towards sustainable trade. At present only LEMIS enables exact details of species imported and their origins and purchasers, and CITES and other UN conventions must interface better between environmental and economic conventions and targets. The lack of efficacy of coverage within CITES is also underscored by the EU Wildlife Trade Regulations, which build on the number of species under-regulation, but also highlights the need for a more comprehensive system globally.

Whilst developing sustainable quotas for offtake are impossible for species with no data on range or populations, better means to monitor and control trade are necessary and could help form the baseline, especially given that over 40% of individuals come from the wild. The cost of enabling the status quo to continue is likely to guarantee the extinction of over-exploited rare, and range-restricted species, especially when the number of species traded annually may be increasing. The drive for rare species entering trade within a few years of description combined with access to more remote areas will expose areas with high endemism to potential exploitation from unsustainable and unmonitored trade, thus better monitoring and reporting standards are needed. Additionally, these naïve populations are vulnerable to pathogens and could potentially replicate the patterns of extinction so far seen in the Americas, and drive significant biodiversity loss. Further regulation, and better monitoring of both wild populations and species and individuals traded is urgently needed to slow the decline of populations and loss of species as a consequence of unsustainable, and largely unmonitored trade in wildlife. This would require databases to monitor international trade of individuals (consistent with not only livestock, but all other commodities) to provide accurate information on what species are being traded, their source, and at what volume. Consistent standards, such as those within LEMIS, provide a blueprint for what could become global wildlife trade databases. LEMIS serves as a framework for agencies wishing to monitor trade; we stress that the data should be fully open and accessible for review and not subject to slow freedom of information requests. For databases to be reliable, central authorities should be delegated at a national level for controlling and certifying traded wildlife, possibly with measures such as DNA barcoding to verify identity, then certify shipments, and be responsible for their export (to prevent laundering). These two approaches would remedy the lack of data, and the potential for laundering, but to prevent trade being unsustainable a shift is needed so that proof of sustainability (i.e., through approved non-detriment findings) is required before trade in a species is allowed. The precautionary principle should become standard practice to ensure that when trading occurs it is based upon a foundation of data to prevent over-exploitation of vulnerable populations; we cannot continue to trade species until we realise that species is already potentially endangered before taking action.

# Materials and methods

## Key resources table

| Reagent type (species) or resource | Designation | Source or reference | Identifiers | Additional information |
|---|---|---|---|---|
| Other | Data S1 – Target Websites Censored.csv | Self-generated via the use of http://www.google.com/ and http://www.bing.com/ | Data S1 | Website review and sampling |
| Other | Data S2. Original AmphibiaWeb data ('AmphibiaWeb 2020-08-29.csv') | AmphibiaWeb: https://amphibiaweb.org/amphib_names.txt | Data S2 | Original AmphibiaWeb Data: Accessed 2020-08-29 |
| Other | Data S3. Snapshot Online Data.csv | Self-generated | Data S3 | Online search results from the contemporary sample |
| Other | Data S4 Temporal Online Data.csv | Self-generated via the Internet Archive's Wayback Machine API and Terraristika (https://www.terraristik.com) | Data S4 | Online search results from the temporal sample |
| Other | Data S5 new_list_amp_jan_FINAL.csv | Self-generated | Data S5 | Species listed purposes from each data source |
| Other | Data S6 supplement_trade_keywords.csv | Self-generated | Data S6 | List of keywords associated the importers and exporters |

*Continued on next page*

*Continued*

| Reagent type (species) or resource | Designation | Source or reference | Identifiers | Additional information |
|---|---|---|---|---|
| Other | Data S7 LEMIS Data AmphiNames.csv | Self-generated by combining aspects of Data S1 and data from LEMIS: Eskew EA, White AM, Ross N, Smith KM, Smith KF, Rodríguez JP, Zambrana-Torrelio C, Karesh WB, Daszak P. 2019. United States LEMIS wildlife trade data curated by EcoHealth Alliance. Zenodo Dataset. doi:10.5281/zenodo.3565869 | Data S7 | Filtered LEMIS data with AmphibiaWeb compatible names: Retrieved using the lemis package: Ross N, Eskew EA, White AM, Zambrana-Torrelio C. 2019. lemis: The LEMIS Wildlife Trade Database. https://github.com/ecohealthalliance/lemis#readme |
| Other | Data S8 Index_of_CITES_ Species_[CUSTOM]_ 2020-09-20 15_51.csv | CITES: http://checklist.cites.org/#/en | Data S8 | Filter CITES appendix data |
| Other | Data S9 gross_ imports_2020-09-20 15_25_comma_ separated.csv | CITES: https://trade.cites.org/# | Data S9 | Filtered CITES data |
| Other | Data S10 Amphibians_ in_trade.csv | Self-generated using aspects of Data S2–S4, S7–S9 | Data S10 | The final dataset |
| Other | Data S11. Amphibians_in_ trade_ METADATA.csv | Self-generated | Data S11 | The final dataset metadata |
| Software, algorithm | R | R Core Team | | Please see appropriate code listed in text |
| Software, algorithm | ArcGis | ESRI | | |
| Other | IUCN species polygons | iucnredlist.org | | |

## Website sampling

We used Google and Bing search engines to discover contemporary websites selling amphibians. We targeted amphibian selling websites in English, French, German, Japanese, Portuguese, and Spanish, to cover the largest herpetofauna pet trade markets. We used appropriately localised versions of the search engines for each language we searched in (Google: https://www.google.com/, https://www.google.fr/, https://www.google.de/, https://www.google.jp/, https://www.google.pt/, https://www.google.es/; Bing: https://www.bing.com/?cc=en, https://www.bing.com/?cc=fr, https://www.bing.com/?cc=de, https://www.bing.com/?cc=jp, https://www.bing.com/?cc=pt, https://www.bing.com/?cc=es). Each localised search engine and language was searched with a Boolean search string:

- English: (amphibians OR frogs OR toads OR salamanders OR newts OR axolotls OR caecilians) AND for sale.
- French: (amphibiens OR grenouilles OR crapauds OR salamandres OR tritons OR axolotls OR céciliens) AND à vendre.
- German: (amphibien OR frösche OR kröten OR salamander OR molche OR axolotls OR caecilian) AND zum verkauf.
- Japanese: (爬虫類 OR カエル OR ウシガエル OR ヒキガエル OR サンショウウオ OR イモリ OR ウーパールーパー OR アシナシイモリ) AND (塒ります OR 販塒).
- Portuguese: (anfíbios OR sapos OR sapos OR salamandras OR tritões OR axolotes OR caecilianos OR rãs OR pererecas) AND à venda.
- Spanish: (anfibios OR ranas OR sapos OR salamandras OR tritones OR axolotls OR cecilias) AND en venta.

We completed the searches in a Firefox private window (*Firefox, 2019*), while signed out of search engine accounts to minimise the impact of previous search history. Our search terms may have missed specialist sellers, specialising in a single genus/species and advertising only with slang.

We downloaded the first 10 pages of search results provided by each search engine (100 URL search results) to produce a list of 200 URLs per language (~1200 URLs overall). We used *assertthat v.0.2.1* (*Wickham, 2019a*), *XML v.3.99.0.3* (*Lang and The CRAN Team, 2018*) and *stringr v.1.4.0* (*Wickham, 2019b*) to extract all URLs present (*Source code 1*). We filtered out URLs associated with internal search engine links, leaving us with a list of potential amphibian selling websites. We simplified the extracted URLs to their base URL (so all URLs ended in. *com,. org,. co.uk*, etc.) and removed duplicates.

We reviewed each website with the goal of determining whether the site sells live amphibians, classifying the type of website (classified ads, commercial, other), determining whether the site explicitly forbade automated data collection, identifying a page within the site to initiate data mining, identifying the most appropriate method of data collection, and identifying any ordering in amphibian listings (the last review goal revealed that websites had a mix of ordering; thereby unlikely to bias results: 21 alphabetically, 17 by featured, 12 by date, 5 by price, 2 by popularity, and 30 whose ordering was unclear). If a website did not sell live amphibians, or explicitly forbade automated data collection, we excluded it. We randomly assigned all accepted websites with a unique ID for further sampling/analysis (*Source data 1*).

The above sampling process was preregistered on 2020-08-29 (osf.io/x5gse). On 2020-09-11, we completed the preregistered sampling and review of 856 websites; we determined that 104 sites would be suitable for searching. However, this was considerably lower than the 151 websites used in previous work (*Marshall et al., 2020*). Therefore, we completed a second search using a simpler search term ('amphibians for sale', and translations) taking the first five pages from both search engines. The new URLs located in the simpler search were reviewed bringing the total reviewed websites to 1069 and the suitable websites to 139 (906 excluded because they did not sell amphibians, 13 specifically stated no automated searching of the website, 6 were duplicates, and the remaining 5 had issues with access).

## Website searching

We used five methods to collect data from websites, applied hierarchically to minimise server load and the number of irrelevant pages searched (*Source code 2*).

### Single page collection

We retrieved a single page, or PDF, for sites that listed the entire stock in a single location. We used the *downloader v.0.4* package (*Chang, 2015*) for the html page retrieval and *pdftools v.2.3.1* (*Ooms, 2019*) to review manually downloaded PDF stock lists.

### Cycling through multi-page lists

When stock lists existed on multiple pages, arranged sequentially (e.g., when a website's internal search functions return 'all amphibians'), we systematically cycled through pages. We identified the maximum search page during website review and ended page cycling when that maximum was reached or the URL returned an error (e.g., 404 error).

### Cycling through multi-page lists, followed by level 1 crawl

If stock lists existed on multiple pages, and the scientific names were only listed behind links on each sequential page, we used the systematically collected pages as a start point for level 1 crawls retrieving all connected pages (i.e., pages holding individual listings or stock details). We used the *Rcrawler v.0.1.9.1* package to perform the crawls (*Khalil, 2018*). We followed the same stop criteria as the basic cycling collection method (method 2).

### Base level 1 crawl

When stock was split between groups, we made use of a level 1 crawl to retrieve all pages (*Khalil, 2018*), setting the page hosting all group links as the start URL.

## Base level 2 crawl

When stock was split into multiple levels of groups, we used a level 2 crawl to collect data at each level (*Khalil, 2018*). For example, stock may be split into 'Frogs' and 'Salamanders', and within 'Frogs' exists links to lists of 'Toads', 'Tree Frogs', and 'Other Frogs'.

For methods including crawling, where possible, we selected keywords in the URL to limit the crawl's scope. For example, all stock may be listed in pages with '/products=frogs/' in the URL. The inclusion of a URL keyword filter prevented us from collecting data from irrelevant pages, while lessening time spent crawling and server load. To further reduce the load placed on servers, we included a 10 s delay between requests. We did not pursue results from websites that actively prevented automated data collection.

In addition to the contemporary sampling of websites, we also sampled for archived web pages originally hosted on Terraristika (https://www.terraristik.com; *Source code 3*). We selected Terraristika to explore the temporal trends in amphibian trade for two reasons: the size of the website and number of species detected in prior contemporary search efforts, and the number of archived web pages available (*Marshall et al., 2020*). We retrieved archive web pages using the Internet Archive's Wayback Machine API (*The Internet Archive, 2013*; *The Internet Archive, 2019*), by adapting code from the *wayback* package (*Rudis, 2017*). We modified the *wayback* code using the *downloader v.0.4* (*Chang, 2015*), *httr v.1.4.2 Wickham, 2018*, *jsonlite v.1.7.0* (*Ooms, 2014*), *lubridate v.1.7.9* (*Grolemund and Wickham, 2011*), and *tibble v.3.0.3* packages (*Müller and Wickham, 2019*).

## Keyword usage

We used species data from AmphibiaWeb as our taxonomic backbone (*AmphibiaWeb, 2020*; https://amphibiaweb.org/amphib_names.txt; accessed 2020-08-29; 2). We created a species list that included all current scientific names and all scientific synonyms. We excluded common names from the keyword list because we did not have common names for all languages nor species, and previous work has shown that common names provide only marginal gains in online data collection efforts (*Marshall et al., 2020*). We also made no attempt to search for partial names or abbreviations (e.g., *Duttaphrynus melanostictus* listed as *D. melanostictus* or *D melanostictus*).

Prior to the keyword search we undertook basic web page text cleaning. We removed all double spaces, special characters, numbers, and html elements, replacing them with single spaces. The basic cleaning meant that genus and species epithets would appear in the same format as the keyword list provided they occur next to each other on the web page. We used *rvest v.0.3.6* (*Wickham, 2019c*), *XML v.3.99.0.3* (*Lang and The CRAN Team, 2018*), and *xml2 v.1.3.2* (*Wickham et al., 2018*) packages to clean and parse the html data.

We used case-insensitive fixed string matching, with *stringr v.1.4.0* package (*Wickham, 2019b*), to search all collected web pages for species names. We used fixed string matching because it has lower computation costs compared with collation matching. Fixed string matching is unable to distinguish between differently coded ligatures or diacritic marks, but our focus on scientific names avoided diacritical marks. Future search efforts using partial or approximate string matching could reveal species we missed if they had only listed with misspelt names or using abbreviations; however, such search efforts would require more computational time, a more thoroughly curated keyword library than what we had access to, and greater caution regarding false positives.

Upon searching a web page for species names, we recorded whether a keyword (species) was present, what accepted species the detected species corresponded to, the page number of the web page, and the website ID (*Source code 4*; *Source data 3* and *4*). We combined final results from the online search with data from LEMIS and CITES (*Source code 5*; retrieved via the R package *lemis v.1.1.0* (*Eskew et al., 2019*; *Eskew et al., 2020*; *Ross et al., 2019*), and https://trade.cites.org/#, respectively).

## Mapping impacts

To understand the dimensions of trade, and how regions may be impacted with different types of trade, we included an additional two data sources (the Mohanty and Measey data based on a collation of published literature, and the IUCN listings of species which state if the species is threatened by trade). We compiled all species on a spreadsheet with the listed purpose from each data source

(*Source data 5*). All species for sale in online stores, we classified as 'pet trade', whereas the Mohanty and Measey data we classified as 'other' and only used these in the total analysis.

For IUCN data the entire list of species listed as 'Use and Trade' for food, medicine, or pets was downloaded. These listings were manually processed and those listing food, medicine, or pets listed, keywords ('food', 'pets', 'medicine') were used to make the process more efficient, but as 'not' was often included in these statements all listings were manually processed, so checking of all listings to verify status was essential. This was used to classify species by use as 'food', 'medicine', 'pharmaceutical', 'pet trade', or 'other uses'. Species for which no form of trade was listed (e.g., 'there is no evidence of trade for this species') were removed from the listings.

For both CITES and LEMIS data, the purpose was collated from the commercially imported listings as well as the personal listings (whilst other categories such as 'research/zoo' were listed directly based on subsets of scientific category data). CITES does not list the importer so only coarse categories listed were usable, whereas for LEMIS keywords could be used for both importers and exporters to determine the likely purpose of the item. Firstly, items were split into 'live' and 'dead'. Companies with dead items were likely to be sourcing items for either meat or pharmaceutical/medicine, whereas live imports could have a variety of purposes, we used a list of keywords associated with the importer and exporter (*Source data 2*) to determine the category each imported item fell into. This still left many items unaccounted for, so as sellers were likely to specialise in one category items were then sorted by seller and other items from that seller listed with the same category. Where a conflict of different listings existed, these were compared to any dead specimens from the same seller, which would indicate that the items were likely to be meat (or medicine/pharmaceuticals). Through this process most items could be sorted to one of the categories, and other suggestive keywords (i.e., 'zoo…' in listings not associated with an actual zoo were classed as pets), and then listings of species traded for each purpose collated in a spreadsheet based on all data sources. Individuals importing species, unless listed for research was also categorised as pets. Whilst there is a degree of uncertainty associated with some of these assigned purposes, it does show that species imported for meat may be a wider selection than realised, as well as those consumed more locally. This was then summed to list the different purposes each species was traded for using LEMIS, and combined with the categories in CITES as well as purposes listed by the IUCN Redlist assessments to produce a list of uses of each species in trade.

For LEMIS summaries of wild capture and captive rearing (*Source code 6* and *7*; *Source data 7*), we filtered the data to only include items that represented single individuals: whole dead animal (LEMIS code = BOD), live eggs (EGL), dead specimen (DEA), live specimen (LIV), specimen (SPE), whole skin (SKI), entire animal trophy (TRO), following the process described in *Hierink et al., 2020*, and *Marshall et al., 2020*. We define non-commercial trade as that termed by LEMIS as: Biomedical research (M), Scientific (S), and Reintroduction/introduction into the wild (Y); whereas captive origin covered Animals bred in captivity (C and F), Commercially bred (D), and Specimens originating from a ranching operation (R); and wild origin only included those listed as Specimens taken from the wild (W). We included all amphibians in origin/purpose summaries, but we only included species detected in LEMIS in final species counts if the full species name listed in LEMIS could be matched to an AmphibiaWeb name or synonym. We relied on LEMIS listing of genus for genera summaries, excluding non-applicable terms (e.g., Non-CITES entry, Anura, Bufonidae, Tadpole).

## Mapping and visualisation

All mapping, bar *Figure 2—figure supplement 2 and 1* (which used on AmphibiaWeb ISOCC country data; *Source code 8*), was completed in ArcMap 10.3. Amphibian data range maps were downloaded from the IUCN (iucnredlist.org) and then species in trade, once corrected for synonyms joined to the shapefile using joins and relates. Individual species maps were then converted into rasters with a resolution of 1 km using the conversion tools. Mosaic to new raster was then used to quantify the species in trade both altogether, or based upon subsets of data such as endangerment, data source (CITES: *Source code 8*, LEMIS: *Source data 7*, Online: *Source data 3* and *4*) or use (pet, meat, research, medicinal/pharmaceutical) to provide global maps depicting each type of pressure.

We also explored temporal trends in CITES, LEMIS, and Online data, plotting changes over time and using a linear regression to account for search effort online (i.e., pages searched; *Source code*

*9*). We also plotted the differences in species lists produced by different languages, and summarised the top 10 most-species rich (by number of unique species) websites' purpose (*Source code 10*).

To calculate the level of coverage on and trade on a national basis, the IUCN maps were intersected with each country to give a country list, and species lacking range maps were compiled to a national level using AmphibiaWeb data. Endangerment and CITES status for species in trade and not traded were associated with this data using the joins and relates function, and quantified using summary statistics before being rejoined to a global map to assay the level of coverage for species in trade at a national level.

### Years of species description

We retrieved all species years of description from the amphibian species of the world database (accessed 2020-10-02; *Frost, 2020*). We used *rvest v.0.3.6* (*Wickham, 2019c*) and *xml2 v.1.3.2* (*Yuan et al., 2018*) to call and retrieve the top search result from the database on a species-by-species basis (each AmphibiaWeb species binomial being used a search term), saving the full character string detailing the species authority (*Source code 10* and *11*). We double-checked the retrieved species authority contained the required species binomial. In cases where species binomial was not included (174), we used *similiars v.0.1.0* (*Sjoberg, 2020*, 2020) to detect minor spelling differences. Ultimately, we found 12 species with non-matching authorities and were detected in the trade; for these 12 species we manually found the appropriate authority. We used LEMIS, CITES (*Source data 9*), and the Online sampling to determine the earliest instance of a species appearing in the trade.

### Software availability

We completed all keyword searches and data review in *R v.3.6.3* (*R Core Development Team, 2020*) and *R Studio v.1.4.669* (*R Studio Team, 2020*). During data manipulation, we also made use of R packages: *dplyr v.1.0.2* (*Wickham et al., 2020*) and *tidyr v.1.1.2* (*Wickham and Henry, 2019*); for data visualisation we used *cowplot v.1.1.0* (*Wilke, 2019*), *ggplot2 v.3.3.2* (*Wickham, 2016*), *ggpubr v.0.4.0* (*Kassambara, 2018*), *ggtext v.0.1.1* (*Wilke, 2020*), *glue v.1.4.2* (*Hester, 2020*), *maps v.3.3.0* (*Becker and Wilks, 2018*), *scico v.1.2.0* (*Pedersen and Crameri, 2018*), and *UpSetR v.1.4.0* (*Gehlenborg, 2019*). We added additional details to the upset plot using *Affinity Designer v.1.8.5.703* (*Serif, 2020*).

We have made code used to search online, filter LEMIS data, generate *Figures 1* and *3–5*, S4, and elements of 6, and retrieve species authorities available at Open Science Framework: https://osf.io/x5gse/?view_only=27109adbb3364dd2b9115752fd912b99. Alongside the code, we have provided all datasheets listed as supplementary material.

## Acknowledgements

We thank the Suranaree University of Technology (School of Biology and the Institute of Science and Institute of Research and Development) for providing the resources required to undertake this research. We thank Inês Silva, Ross Creelman, and Akihiro Nakamura for checking translations of search terms. Funding Chinese National Natural Science Foundation (Grant #:U1602265, Mapping Karst Biodiversity in Yunnan). Supported by the High-End Foreign Experts Program of Yunnan Province (Grant #:Y9YN021B01, Yunnan Bioacoustic monitoring program). Supported by the CAS 135 program (No. 2017XTBG-T03).

## Additional information

### Funding

| Funder | Grant reference number | Author |
|---|---|---|
| National Science Foundation of China | U1602265 | Alice C Hughes |
| Chinese Academy of Sciences | XDA20050202 | Alice C Hughes |
| Office of International Affairs | Y9YN021B01 | Alice C Hughes |
| Chinese Academy of Sciences | 2017XTBG-T03 | Alice C Hughes |

Chinese Academy of Sciences    Y4ZK111B01              Alice C Hughes

The funders had no role in study design, data collection and interpretation, or the decision to submit the work for publication.

## Author contributions

Alice C Hughes, Conceptualization, Resources, Formal analysis, Supervision, Funding acquisition, Methodology, Writing - original draft, Writing - review and editing; Benjamin Michael Marshall, Software, Formal analysis, Methodology, Writing - original draft, Writing - review and editing; Colin T Strine, Methodology, Writing - original draft, Writing - review and editing

## Author ORCIDs

Alice C Hughes https://orcid.org/0000-0002-4899-3158
Benjamin Michael Marshall http://orcid.org/0000-0001-9554-0605

## Decision letter and Author response

Decision letter https://doi.org/10.7554/eLife.70086.sa1
Author response https://doi.org/10.7554/eLife.70086.sa2

# Additional files

## Supplementary files

- Source code 1. Code used to extract URLs from saved search result pages.

- Source code 2. Code to collect website data using the hierarchical search method.

- Source code 3. Code to collect website data from the wayback machine.

- Source code 4. Code used to implement string matching searches for species keywords.

- Source code 5. Code used to compile website search results with Law Enforcement Management Information System (LEMIS) and Convention on International Trade in Endangered Species (CITES) data.

- Source code 6. Code used to filter initial Law Enforcement Management Information System (LEMIS) data.

- Source code 7. Code used to summarise and explore Law Enforcement Management Information System (LEMIS) data.

- Source code 8. Code used to generate summary figures.

- Source code 9. Code used to generate figures showing change over time.

- Source code 10. Code used to plot the different species counts between languages used during online searches.

- Source code 11. Code used to retrieve species authorities.

- Source code 12. Code used to calculate and plot lag times between species description and appearance in the trade.

- Source data 1. Website review and sampling ('Target Websites Censored.csv').

- Source data 2. Original AmphibiaWeb data ('AmphibiaWeb 2020-08-29.csv').

- Source data 3. Online search results from the contemporary sample ('Snapshot Online Data.csv').

- Source data 4. Online search results from the temporal sample ('Temporal Online Data.csv').

- Source data 5. Species listed purposes from each data source ('new_list_amp_jan_FINAL.csv').

- Source data 6. List of keywords associated the importers and exporters ('supplement_trade_keywords.csv').

- Source data 7. Filtered Law Enforcement Management Information System (LEMIS) data with AmphibiaWeb compatible names ('LEMIS Data AmphiNames.csv').

- Source data 8. Filter Convention on International Trade in Endangered Species (CITES) appendix data ('Index_of_CITES_Species_[CUSTOM]_2020-09-20 15_51.csv').

- Source data 9. Filtered Convention on International Trade in Endangered Species (CITES) data ('gross_imports_2020-09-20 15_25_comma_separated.csv').
- Source data 10. The final dataset ('Amphibians_in_trade.csv').
- Source data 11. The final dataset metadata ('Amphibians_in_trade_METADATA.csv').
- Transparent reporting form

## Data availability

All code and data is available in Supplements. All data used is also provided in it's used form and listed in the Key Resources table.

The following datasets were generated:

| Author(s) | Year | Dataset title | Dataset URL | Database and Identifier |
|---|---|---|---|---|
| Hughes AC | 2020 | CITES Database | https://trade.cites.org | CITES, Database |
| CITES | 2020 | CITES Checklist | http://checklist.cites.org/#/en | CITES, Checklist |
| Eskew EA, White AM, Ross N, Smith KM, Smith KF, Rodríguez JP, Daszak P | 2019 | United States LEMIS wildlife trade data curated by EcoHealth Alliance | https://doi.org/10.5281/zenodo.3565869 | Zenodo, 10.5281/zenodo.3565869 |
| IUCN | 2020 | IUCN Redlist of Threatened species | https://www.iucnredlist.org/search#link | IUCN 2020, Redlist |

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
