## [Decision Letter]

**Acceptance summary:**

These are pressing times for nature, which stands alone multiple impacts of human-fuelled ecological stressors. Hughes et al. increase dramatically the understanding of one of them, the legal and illegal trade of amphibians. To do so, they propose a wider set of computer methods and web searches that we are sure will inspire conservation biologists and ecologists interested in the trade of amphibians and well as other wildlife taxa.

**Decision letter after peer review:**

Thank you for submitting your article "Gaps in global wildlife trade monitoring leave amphibians vulnerable" for consideration by *eLife*. Your article has been reviewed by 2 peer reviewers, and the evaluation has been overseen by a Reviewing Editor and George Perry as the Senior Editor. The following individual involved in review of your submission has agreed to reveal their identity: Mark Auliya (Reviewer #2).

Essential revisions:

Please notice that while most questions for you to address are minor, and while most likely the next version will not go through additional external review, we expect you to provide us with a revised version and point-by-point response to all criticism below.

*Reviewer #2:*

By linking several databases, the authors tried to measure the impact of trade on all amphibian species. Thereby origins of species traded, volumes, the purpose for trade have been assessed while noting that several loopholes exist in making overall robust assessments e.g., dynamics in trade and taxonomy. However, indicating that gaps and shortcomings are based on the way current databases available have been set up, the authors did an enormous job in applying the extensive digital methods to achieve the best possible reflection of the current amphibian trade. Through the use of several software programmes to measure/analyse current databases, also figures could be built to visualize e.g. trends. This is one of the major strengths of this paper. With the various specifically named methodological queries as well as the use of categorized keywords, which were essential for the inclusion and linking of the various databases, unambiguous results could be generated in the best possible sense, on the basis of which correct and convincing recommendations were made. To make it explicitly clear again, due to the lack of data on the global anthropogenic use of amphibian species, this seemingly complex applied methodological approach was necessary to shed light on the dark. In return, the effort would have been many times less, there would have been more comprehensive and informative databases that transparently and up-to-date illuminate the population status of species, their threats and the impact of trade. The importance of this work is essential to understand how much the various interest groups are lagging behind in order to communicate responsibly and transparently the use of resources, in this case the most threatened group of vertebrates, amphibians; thus, it is difficult to understand as the reader learns here how easy it is to trade species that have already been classified as threatened.

If I would have to mention weaknesses of this paper there are none that I would address explicitly. Apart from the comprehensive Suppl. Mat., I would just not overload the actual manuscript with figures and make sure that these are self-explanatory. As already mentioned, the methodological part in particular is very extensive and complex, but this is essential for this type of study.

I only few suggestions/recommendations. Make sure you standardize some of the terms used, especially the IUCN red List categories. Also make sure that there is only one "database" as such noted, that refers to CITES. There is no IUCN trade database for example, of course I know what you meant. I have included quite a few comments in the pdf for you to check/evaluate, I also proposed few more references for your perusal.

Two things should be included, I think. A sentence or two regarding mortalities in trade (x %), prior to export and during intercontinental shipments, as especially amphibians are prone to die off during temporary captive and transport conditions; I have provided one reference for you to check. Also, the impact taxonomy/systematics have on species informing a wide readership (even wider due to online access) on species discoveries; here revised policies need to be implemented; there are quite a few papers that refrain naming localities of new taxa described, e.g., in orchids, cacti, reptiles, to hinder unscrupulous collectors.

*Reviewer #3:*

While it is wildly assumed that the trade in wildlife is well documented and data are thorough and widely available, this is not the case. The authors scour online sources (databases, websites, marketplaces), in multiple languages, to assess the true extent of wildlife trade related to those values reported and find large discrepancies. Wildlife trade has both direct and indirect effects on wild populations of amphibian species and therefore having more accurate values is essential for measuring potential effects. They call for change in how data are collected and reported so that those data can properly influence policy and conservation measures.

I'd like to congratulate the authors on an interesting and seemingly thorough paper. I was unfamiliar with these methods and spent a fair bit of time trying to truly understand how their code works, and I think those methods/code could help broader studies, so I thank them also for their transparency. The goals of the paper are well sought out and achieved, but it's quite a descriptive paper.

Find the use of Canary in the coalmine used twice to be a little hand-wavy as this paper is dealing with pure numbers and the extent of wildlife trade, not directly how this is affecting wild populations. I completely agree with the statement, but using it twice in key areas (e.g. start of the discussion) is distracting.

There are also some additional citations (perhaps lesser-known) which could be useful given the taxonomic breadth and the underreporting for South America. For example this popular article on poison frog smuggling:

https://www.dendrobates.org/2007/01/rare-species-for-sale-the-smuggling-crisis/

There's also this article on biocommerce in poison frogs with comments on numbers reported which may additionally have useful references within:

https://www.sciencedirect.com/science/article/abs/pii/S1617138120300492?fbclid=IwAR2wG7Mre0OwJUfhxwfhNWhS0IdC0gqn4OgmqRQMMC-TNqovSo6jJX8LphM

I would also like to see the authors propose (albeit briefly) what their suggestions are for data to be collected, by whom, where it would be available, etc. to pitch best practices. They repeatedly call for those data to exist and be used, so why not describe your concept of how/when those data could collected? Surely this can not be said enough, and given the authors have thoroughly dug for such data they will have useful suggestions.

---

## [Author Response]

Reviewer #2:I only few suggestions/recommendations. Make sure you standardize some of the terms used, especially the IUCN red List categories. Also make sure that there is only one "database" as such noted, that refers to CITES. There is no IUCN trade database for example, of course I know what you meant. I have included quite a few comments in the pdf for you to check/evaluate, I also proposed few more references for your perusal.Two things should be included, I think. A sentence or two regarding mortalities in trade (x %), prior to export and during intercontinental shipments, as especially amphibians are prone to die off during temporary captive and transport conditions; I have provided one reference for you to check.

Thanks, a line has been added:

“Furthermore, quantitative analysis of the volumes of species in trade often relies on import data (i.e. LEMIS) and ignores mortality during transit and transport, which has been shown to be as high as 72% in some studies (Ashley et al., 2014), with mortality in amphibians higher than all other groups (45% within 10 days of confiscation). Such statistics are alarming, and also highlight that the number of animals exported may be far higher than the anticipated demand to compensate for mortality before sale.”

Also, the impact taxonomy/systematics have on species informing a wide readership (even wider due to online access) on species discoveries; here revised policies need to be implemented; there are quite a few papers that refrain naming localities of new taxa described, e.g., in orchids, cacti, reptiles, to hinder unscrupulous collectors.

We totally agree, we have added the following:

“and even scientific descriptions of species have been found to enable these newly described species to be targeted for trade (Yang et al., 2015; Yeager and Zarling 2020).”

Reviewer #3:While it is wildly assumed that the trade in wildlife is well documented and data are thorough and widely available, this is not the case. The authors scour online sources (databases, websites, marketplaces), in multiple languages, to assess the true extent of wildlife trade related to those values reported and find large discrepancies. Wildlife trade has both direct and indirect effects on wild populations of amphibian species and therefore having more accurate values is essential for measuring potential effects. They call for change in how data are collected and reported so that those data can properly influence policy and conservation measures.I'd like to congratulate the authors on an interesting and seemingly thorough paper. I was unfamiliar with these methods and spent a fair bit of time trying to truly understand how their code works, and I think those methods/code could help broader studies, so I thank them also for their transparency. The goals of the paper are well sought out and achieved, but it's quite a descriptive paper.

Thank you, we appreciate that and hope that our approaches can provide a more coherent understanding of trade for other taxa, or languages.

Find the use of Canary in the coalmine used twice to be a little hand-wavy as this paper is dealing with pure numbers and the extent of wildlife trade, not directly how this is affecting wild populations. I completely agree with the statement, but using it twice in key areas (e.g. start of the discussion) is distracting.

Thank you, we have rephrased the second as:

“Amphibian declines are often considered to provide an early warning of potential declines in other taxa”.

There are also some additional citations (perhaps lesser-known) which could be useful given the taxonomic breadth and the underreporting for South America. For example this popular article on poison frog smuggling:https://www.dendrobates.org/2007/01/rare-species-for-sale-the-smuggling-crisis/

Thank you, this is a useful and sad example. It is challenging to reference given that it is not an article, but will be useful to mention during press-releases to illustrate some of the issues discussed.

There's also this article on biocommerce in poison frogs with comments on numbers reported which may additionally have useful references within:https://www.sciencedirect.com/science/article/abs/pii/S1617138120300492?fbclid=IwAR2wG7Mre0OwJUfhxwfhNWhS0IdC0gqn4OgmqRQMMC-TNqovSo6jJX8LphM

Thank you, this has some useful examples, and has now been cited in text. We also use this reference to discuss more sustainable alternatives for trade.

I would also like to see the authors propose (albeit briefly) what their suggestions are for data to be collected, by whom, where it would be available, etc. to pitch best practices. They repeatedly call for those data to exist and be used, so why not describe your concept of how/when those data could collected? Surely this can not be said enough, and given the authors have thoroughly dug for such data they will have useful suggestions.

Thank you, we agree. We have added the following:

“This would require databases to monitor international trade of individuals (consistent with not only livestock, but all other commodities) to provide accurate information on what species are being traded, their source and at what volume. Consistent standards, such as those within LEMIS provide a blueprint for what could become global wildlife trade databases. LEMIS serves as a framework for agencies wishing to monitor trade; we stress that the data should be fully open and accessible for review and not subject to slow freedom of information (FOI) requests. For databases to be reliable, central authorities should be delegated at a national level for controlling and certifying traded wildlife, possibly with measures such as DNA barcoding to verify identity, then certify shipments and be responsible for their export (to prevent laundering). These two approaches would remedy the lack of data, and the potential for laundering, but to prevent trade being unsustainable a shift is needed so that proof of sustainability (i.e. through approved non-detriment findings) are required before trade in a species is allowed.”